# Inhibition of Th2 Differentiation Accelerates Chronic Wound Healing by Facilitating Lymphangiogenesis

**DOI:** 10.3390/biomedicines13051026

**Published:** 2025-04-24

**Authors:** Bracha L. Pollack, Jeremy S. Torrisi, Geoffrey E. Hespe, Gopika Ashokan, Jinyeon Shin, Babak J. Mehrara, Raghu P. Kataru

**Affiliations:** Plastic and Reconstructive Surgery Service, Department of Surgery, Memorial Sloan Kettering Cancer Center, New York, NY 10065, USA; pollackb@mskcc.org (B.L.P.); torrisi.jeremy@gmail.com (J.S.T.); hespeg@mskcc.org (G.E.H.); ashokang@mskcc.org (G.A.); shinj1@mskcc.org (J.S.); mehrarab@mskcc.org (B.J.M.)

**Keywords:** chronic wounds, wound healing, lymphangiogenesis, lymphedema, lymphatics, LECs, Th2 inflammation

## Abstract

**Background/Objectives**: Chronic wounds pose a significant healthcare burden, and there remains no effective animal model for study. We aimed to develop a mouse model of chronic wounds that remain open for at least 4 weeks and to investigate the role of the lymphatic system in wound healing. **Methods**: Full-thickness excisional wounds were created on the dorsal surface of mouse tails to simulate chronic wounds. Lymphatic drainage was assessed using FITC–dextran lymphangiography. Histology and immunofluorescence were used to analyze immune cell infiltration. The effect of inhibiting Th2 differentiation via IL-4 and IL-13 neutralization on wound closure was also evaluated. **Results**: Our chronic wound model was successful, and wounds remained open for 4 weeks. Impaired lymphatic drainage was observed extending beyond the wound area. Increased CD4+ T-helper cell infiltration and Th2 cell accumulation were observed in the impaired lymphatic drainage zone. Inhibition of IL-4 and IL-13 accelerated wound healing. **Conclusions**: Impaired lymphatic drainage and Th2-mediated inflammation contribute to delayed healing, and gradients of lymphatic fluid flow are associated with spatial differences in lymphangiogenesis. Targeting Th2 cytokines may offer a novel therapeutic approach for chronic wounds.

## 1. Introduction

Chronic wounds—those that fail to close spontaneously in 6 weeks or more—are a major source of morbidity and biomedical expenditures afflicting approximately 6 million patients at a cost of USD > 9 billion annually [1,2,3,4,5,6]. One of the challenges in studying the pathophysiology of chronic wounds is the difficulty in developing animal models in which wounds fail to close spontaneously [7,8,9]. Commonly used animal models, such as rodents or rabbits, have highly pliable skin with an underlying layer of muscle that promotes rapid wound contracture and closure even after the excision of large patches of skin [10]. As a result, researchers have used external beam radiation [11,12,13], induction of tissue ischemia [14,15,16,17], and metabolic abnormalities such as hyperglycemia or diabetes [16,18,19,20,21,22,23], to delay wound healing. However, these interventions may have independent effects on wound healing that may limit translational potential. Other investigators employed full-thickness murine wounds that are stented with an external device to prevent wound closure by contracture [16,18,19,20]. These models are useful; however, in healthy animals, these wounds still close in 14–21 days. Thus, developing a rodent model in which wounds remain open for long periods of time (4 weeks or more) is important and may have translational relevance.

Previous studies have largely focused on the role of the angiogenesis for the treatment of chronic wounds given that wound ischemia is a major cause of chronic wound development [24,25,26,27]. The lymphatic system is a key regulator of immune cell trafficking and the clearance of cellular debris, two processes that are important for chronic wound healing to occur [28,29,30,31]. However, the contribution of this system to wound healing remains understudied. Therefore, the goal of this current study was twofold: (1) develop a mouse model of chronic wounds that fails to close spontaneously for 4 weeks or more without additional interventions and (2) analyze the changes in lymphatic vessels drainage and function in the regions surrounding the open wound.

Mouse tail skin, unlike skin elsewhere in the mouse body, does not have an underlying layer of muscle and, like glabrous skin in primates, has dense collagen attachments to the underlying intrinsic tail musculature and bone. We hypothesized that these connections prevent or delay wound closure by contraction. We modified a previously described mouse tail excisional skin model of lymphedema [32,33] and showed that these wounds remain open for approximately 4 weeks and have impaired lymphatic drainage wider than the area of the open wound. We also demonstrated that the area of impaired lymphatic drainage corresponds spatially to the increased infiltration of CD4+ T-helper cells and accumulation of T-helper-2 (Th2) cells. Finally, we showed that the inhibition of naïve T-helper cell differentiation to Th2 cells, a process dependent on interleukin 4 (IL4) and IL13, significantly accelerates wound healing.

## 2. Materials and Methods

**Mice and wound model:** All experimental protocols were reviewed and approved by the Institutional Animal Care and Use Committee (IACUC) at Memorial Sloan Kettering Cancer Center (MSKCC). Female 8–10-week-old C57BL/6J mice (Jackson Laboratories, Bar Harbor, ME, USA) were used for all experiments. We created a 3 × 3 mm full-thickness excisional skin wound on the dorsal surface of the mouse tail (i.e., half the circumference of the tail) beginning at a point that was located 2 cm from the base of tail. The deep lymphatics—located adjacent to the lateral tail veins on each side of the tail—were visualized but not ligated during the creation of the full-thickness wound. Likewise, the ventral skin of the tail was kept intact to maintain integrity of the capillary lymphatics in the skin. After surgery, mice were placed in a recovery chamber, kept warm, and allowed to recover in single cages. The wounds were left open and allowed to granulate closed.

**Histology:** For histological studies, mice were sacrificed at two-week intervals up to 6 weeks postoperatively (*n* = 5 for each time point). Prior to sacrifice, tail wounds were photographed in a standardized manner using a tripod to enable calculation of the area of the wound. The area that remained open from proximal end to distal end at each time point was analyzed using Fiji software (Fiji software v1.54p; NIH, Bethesda, MD, USA). Wound histology was performed by harvesting the tail 1 cm both distal and proximal to the observable wound bed. Tail sections were fixed in 4% paraformaldehyde solution (Sigma-Aldrich, St. Louis, MO, USA) at 4 °C for 4 h. After fixation, tissues were decalcified in EDTA (Santa Cruz Biotechnology, Santa Cruz, CA, USA) for 1 week. The tails were then sectioned longitudinally through the wound sites and embedded in paraffin. Histology sections of 5 µm were then obtained using a rotary microtome (Leica Microsystems, Inc., Buffalo Grove, IL, USA). Hematoxylin (Dako, Carpinteria, CA, USA) and eosin (Sigma-Aldrich, St. Louis, MO, USA) staining was performed using standard techniques.

**Fluorescent immunohistology:** Immunofluorescent staining was performed using our previously described methods. Briefly, paraffin-embedded tissues were rehydrated and subjected to antigen unmasking using boiling sodium citrate (Sigma-Aldrich). Following antigen unmasking, endogenous peroxidase activity was quenched using 3% H_2_O_2_ (Sigma-Aldrich). Non-specific antibody binding was blocked using 2% bovine serum albumin/10% secondary-animal serum solutions. Tissues were incubated with primary antibody overnight at 4 °C. Primary antibodies used were anti-lymphatic vessel endothelial hyaluronan receptor (LYVE)-1 (ab14917; Abcam, Cambridge, MA, USA), anti-CD4 (AF554; R&D Systems, Minneapolis, MN, USA), and anti-Gata3 (sc-22206; Santa Cruz Biotechnology). Secondary immunofluorescent antibodies used were donkey anti-rabbit and donkey anti-goat (A-21206 and A-11058, respectively; both from Life Technologies, Grand Island, NY, USA).

Antibody staining was visualized with fluorescent-conjugated secondary antibodies, and slides were mounted with Mowiol (Sigma-Aldrich). Immunofluorescent sections were scanned using a Mirax slide scanner (Zeiss, Munich, Germany). The peri-wound area—defined as areas within 3 mm either distal or proximal to the wound bed—was analyzed, and cell counts were performed in 40× high-power fields (HPFs) using Slide Viewer (3DHistech, Budapest, Hungary) with a minimum of 5 animals/group and 1–2 HPF/animal. Analysis of lymphatic vessel density was performed in a similar manner using 20× HPF views. Cell counts and analysis of lymphatic vessel density were performed by two blinded reviewers.

**Neutralizing antibody treatment:** For neutralizing antibody-treatment experiments, mice underwent the previously described tail-excision wound protocol and were treated with isotype antibody (Bio-X-Cell, West Lebanon, NH, USA), anti-IL-4 mAb (clone 11B11, 5g/g/dose administered intraperitoneally weekly; Bio-X-Cell), or anti-IL-13 mAb (clone 38213, 5g/g/dose administered intraperitoneally every 4 days; R&D Systems, Minneapolis, MN, USA) [34,35] starting on post-operative day 3. Mice were sacrificed 3 weeks post-operatively. Analysis of wound closure was performed as previously described.

**Fluorescent micro-lymphangiography:** Fluorescent micro-lymphangiography is a commonly used technique to visualize lymphatic vessels and drainage in vivo. This procedure was performed weekly following surgery with fluorescein isothiocyanate (FITC)-conjugated with dextran (2000 kDa, 10 mg/mL; Invitrogen, Carlsbad, CA, USA) using our previously published methods [36,37,38,39]. In this technique, the large molecular weight of the dextran-conjugated FITC molecule limits the absorption of the fluorescent marker to the lymphatic system, enabling the visualization of capillary lymphatics that have a characteristic honeycomb appearance. FITC dextran (10 µL) was injected into the distal tip of the tail (i.e., far away from the zone of injury), and the lymphatic channels around the wound were visualized 10 min later using a Zeiss V12 Stereolumar microscope (Caliper Life Sciences, Hopington, MA, USA). Brightfield images of the same frame were also captured. The area around the wound that was devoid of FITC+ lymphatic vessels (i.e., dysfunctional lymphatics) was measured by drawing a region of interest along the border of the FITC+ region and calculating the area of this area using computer software (Fiji v1.54p). This area was compared with the size of the open wound on brightfield images using the same methods. A minimum of 5 animals were evaluated at each time point.

**Flow cytometry of tail skin:** Single-cell suspensions of tail skin tissue were prepared by mechanical dissociation and incubated with digestion buffer containing collagenase D, DNAse I, and Dispase II (Roche Diagnostics; Indianapolis, IN, USA). Erythrocytes were lysed with RBC lysis buffer (eBioscience; San Diego, CA, USA). Single-cell suspension samples were stained with different combinations of the following fluorophore-conjugated mouse monoclonal antibodies: BV605-conjugated anti-CD45(30-F11, 103139), PerCP-conjugated anti-CD4 (RM4-5, 100540), FITC-conjugated anti-CD3 (17A2,11-0032-82), PE/Cy7-conjugated anti-CCR4 (2G12, 131213), APC-conjugated anti-CCR8 (SA214G2, 150310), PE-conjugated anti-Podoplanin (8.1.1, no. 127407), and APC-conjugated anti-CD31 (MEC 13.3 no. 102509). Non-specific staining was reduced with Fc receptor block (rat monoclonal anti-CD16/32, no. 14-0165-85, eBiosciences, San Diego, CA, USA). DAPI viability dye was also used to exclude dead cells. Single-stain compensation samples were created using UltraComp eBeads (#01-2222-42; Affymetrix, Inc., Santa Clara, CA, USA). Samples were fixed in 25% PFA with PBS overnight and analyzed via flow cytometry the following morning. Flow cytometry was performed using a BD Fortessa flow cytometry analyzer (BD Biosciences; San Jose, CA, USA), and data were analyzed with FlowJo software (v10.10.0, Tree Star; Ashland, OR, USA).

**Statistics:** Statistical analyses were performed using GraphPad Prism (GraphPad Software v10.0.3; San Diego, CA, USA). The normal distribution of all the data sets was checked using the Shapiro–Wilk normality test. One-way ANOVA was used to compare differences between two groups, and two-way ANOVA or the Kruskal–Wallis test was used to compare differences between three groups. Data are presented as mean ± standard deviation unless otherwise noted, and *p* < 0.05 was considered significant. In figures, * denotes *p* < 0.05, ** denotes *p* < 0.01, *** denotes *p* < 0.001, and **** or # denotes *p* < 0.0001.

## 3. Results

### 3.1. Full-Thickness Tail Excisions Result in Chronic Wounds That Persist for 4 Weeks and Are Surrounded by a Zone of Dysfunctional Lymphatics

Quantification of the wound area showed that most tail wounds healed by 4 weeks following surgery (Figure 1A–C), with all wounds fully closing within 5 weeks. Histological analysis of the wounds showed a large area of granulation tissue underneath the wound area extending to the intrinsic tail musculature; this area peaked two weeks following surgery and decreased gradually thereafter (Figure 1D,E).

We next sought to determine whether chronic wounds have impaired lymphatic drainage using microlymphangiography using dextran-conjugated FITC. We found nearly identical patterns of fluorescent uptake—as characterized by a honeycomb appearance of the tail capillary lymphatics—in the skin lymphatics located either proximal or distal to the excision site (Figure 2A). However, we noted the presence of a small amount of dye in the wound at the 1- and 2-week time points resulting from leakage from the transected dermal lymphatics. We also observed a higher intensity of fluorescent staining in the lymphatic capillaries located distal to the tail skin excision site, likely reflecting the direction of lymphatic flow and spatial patterns of impaired lymphatic drainage. Interestingly, we noted that the zone of abnormal lymphatic function, as represented by an absence of FITC staining in the skin capillary lymphatics, extends well beyond the area of the excisional wound at all time points evaluated (compare red and blue outlines in Figure 2A–C). Abnormal lymphatic function persisted even 6 weeks after surgery—as evidenced by a ring of diminished FITC uptake and absence of lymphatic vessels that crossed the excisional site—suggesting that lymphatic regeneration and remodeling lags behind wound closure.

The visualization of capillary lymphatics with LYVE-1 staining in histological sections demonstrated that compared with normal skin, there is a considerable lymphangiogenic response in the regions surrounding the open wound (Figure 2D,E). Newly formed lymphatic capillaries were also noted in the wound bed; however, these lymphatic vessels appeared in the wound at later time points, suggesting that lymphangiogenesis begins at the wound periphery and extends to the wound bed. Taken together, our findings suggest that the lymphatics surrounding wounds have impaired function, that these impairments persist even when the wound is completely re-epithelialized, and that gradients of lymphatic fluid flow (i.e., distal to proximal) are associated with spatial differences in lymphangiogenesis following wounding. In addition, the presence of lymphatic vessels in the surrounding regions but no uptake of FITC suggest that some of these newly formed lymphatic vessels are dysfunctional and incapable of transporting interstitial fluid.

### 3.2. Chronic Wounds Result in a T-Helper Cell Infiltration and Th2 Differentiation

Previous studies have shown that lymphedema and impaired lymphatic drainage are associated with a mixed Th1/Th2 chronic inflammatory response and that Th2 cells contribute to the pathology of lymphedema by promoting tissue fibrosis, impairing lymphatic proliferation, and either directly or indirectly decreasing interstitial fluid clearance [36,40,41,42]. The immunofluorescent staining of mouse tail wound tissues demonstrated a marked increase in the number of CD4+ T-helper cells in the dermis and subcutaneous tissues as compared with normal skin beginning at the earliest time point we analyzed (2 weeks) and peaking by 4 weeks (Figure 3A,B). Although the number of CD4+ cells per HPF decreased somewhat 6 weeks after surgery, this number was still significantly higher than normal tail skin, suggesting that CD4+ inflammatory responses persist even after wounds are completely re-epithelialized.

The differentiation of naïve CD4+ cells to Th2 polarized cells requires the expression of the transcription factor GATA3 and stimulation by IL4 and IL13. We therefore analyzed Gata3 expression as a surrogate for Th2 differentiation and found that similar to the patterns of staining of CD4+ cells, the number of Gata3+ cells in the dermis and subcutaneous tissues increased significantly postoperatively peaking by 4 weeks after surgery (Figure 3C,D). The number of Gata3+ cells/HPF also remained increased even 6 weeks after surgery. This was further analyzed and confirmed via flow cytometry in which we quantified CD4+ T cells as a percentage of total CD45+ cells, and specifically Th2 cells (CCR4+ and CCR8+ T cells), and found that the quantity peaked at 4 weeks post-operatively. (Figure 3E–H).

### 3.3. Healing of Excisional Wounds and Lymphangiogenesis Is Accelerated by Inhibition of Th2 Cytokines

We have previously shown that the inhibition of Th2 differentiation using IL-4- or IL-13-neutralizing antibodies is an effective means of treating lymphedema and improving lymphatic function [36]. We therefore sought to determine whether the inhibition of Th2 differentiation could accelerate wound repair in the mouse tail. To do this, beginning on the day of surgery and for 3 weeks postoperatively, we treated mice with excisional tail wounds with control (same isotope but non-specific monoclonal antibodies (mAbs)) or neutralizing monoclonal antibodies targeting either IL-4 or IL-13. Interestingly, we found that mice treated with either IL4 or IL13 mAbs had significantly faster re-epithelialization of their wounds as evidenced by gross observation and histological evaluation (Figure 4A,C). IL4 and IL13 mAb had wounds that healed faster than isotype controls at weeks 2 and 3 post-operation (Figure 4B). IL4 and IL13 mAb-treated animals also had less granulation tissue in the wound bed at the 6-week time point, suggesting that this treatment decreased inflammation. (Figure 4D). This hypothesis was confirmed with immunofluorescent staining demonstrating a more than 40% decrease in the number of CD4+ cells/HPF as compared with controls (Figure 5A,B). Consistent with this finding, we noted a decrease in the number of Gata3+ cells in the dermis and subcutaneous tissues of the wound edges (Figure 5C,D). In addition, both CD4+ T cells and, more specifically, Th2+ T cells were significantly decreased at the 3-week time point in the IL-4mAB and IL-13mAB groups (Figure 5E–H).

IL4 and IL13 have direct effects on lymphatic endothelial cells (LECs) and inhibit cellular proliferation, differentiation, and lymphangiogenesis [43,44]. Consistent with these previous studies, we found that the inhibition of IL4 or IL13 with neutralizing antibodies significantly increased dermal lymphatic vessel density and lymphangiogenesis as reflected by an increased number of LYVE-1+ cells/HPF (Figure 6A,B). In addition, the number of LECs increased in the IL-4 mAb-treated group (Figure 6C,D). But we did not note any significant changes in angiogenesis or CD31+ vascular density based on fluorescent immunohistology or flowcytometry quantification (Appendix A). Taken together, these findings suggest that the inhibition of Th2 differentiation can decrease chronic inflammatory reactions in excisional wounds and that this treatment may accelerate wound re-epithelialization and promote lymphangiogenesis.

## 4. Discussion

The pathogenesis of chronic wounds is a complex and multifactorial process in which venous insufficiency, tissue hypoxia, and bacterial colonization have been extensively characterized as contributing factors [22,45]. In contrast, the role of lymphatic dysfunction in wound healing is poorly understood and newly emerging. In the current study, we show that lymphangiogenesis initiates at the wound edges beginning approximately 2 weeks after surgery and that newly formed lymphatic vessels cross into the wound bed at later stages in wound healing corresponding to re-epithelialization. In addition, we found that the inhibition of Th2 cytokines, IL4 or IL13, significantly increased lymphangiogenesis and was associated with more rapid wound closure. These findings are consistent with previous reports demonstrating that Th2 cytokines have potent anti-lymphangiogenic properties and act directly on lymphatic endothelial cells to decrease cellular proliferation, migration, and differentiation [36,43].

Several lines of evidence support the hypothesis that lymphatic regeneration is associated with improved wound healing [46]. For example, analysis of clinical biopsy specimens treated with vacuum-assisted closure devices demonstrated that increased lymphatic vessel density was correlated with more rapid wound healing [47]. This finding is consistent with other histological studies demonstrating that lymphangiogenesis occurs concomitantly with angiogenesis in the normal wound healing process [48,49,50] and that states such as venous insufficiency, which can exacerbate chronic wounds, coincide with lymphatic dysfunction [51]. Similarly, Martínez-Corral et al. showed that treatment with steroids not only delayed wound closure but also potently decreased lymphangiogenesis [52]. Dysfunctional lymphatics and impaired lymphangiogenesis are also thought to play a role in diabetic chronic wounds [53,54]; in these wounds, the formation of lymphatic vessels is a rate-limiting step for wound healing, and this process is dependent on the elaboration of lymphangiogenic growth factors from activated macrophages [55]. Consistent with these findings, other studies have shown that the adenoviral delivery of vascular endothelial growth factor C (VEGF-C), a potent lymphangiogenic growth factor, accelerates skin wound healing in diabetic mouse models [56,57]. Thus, there is significant evidence that the lymphatic system plays a crucial role in wound repair and that treatments that directly induce lymphangiogenesis (i.e., delivery of lymphangiogenic growth factors) or ones that decrease inhibitory signals for lymphatic repair (e.g., blockade of Th2 cytokines) may have a role in treating chronic wounds that are recalcitrant to other treatment modalities.

Although lymphangiogenesis and wound healing are correlated, the cellular mechanisms that regulate this interaction are less well understood. Indeed, the lymphatic system plays a crucial role in a number of biologic processes—the clearance of interstitial fluid and cellular debris, trafficking of leukocytes, and regulation of immune reactions—that may regulate wound repair [58,59]. For example, a major function of the lymphatic system in healthy skin is to clear interstitial fluid; thus, it is possible that lymphatic dysfunction in chronic wounds or during the early phases of wound repair results in the accumulation of interstitial fluid, thereby increasing local tissue swelling and increasing the distance between cells and their blood supply with resultant gradients of hypoxia [60,61,62].

Failure to clear cellular debris by dysfunctional lymphatics may also promote chronic inflammatory reactions that impede wound repair and regeneration [63]. This hypothesis is supported by the efficacy of compression devices in the treatment of chronic venous ulcers—chronic wounds in which tissue edema is known to play a key role in pathogenesis [64,65]. Functional lymphatics may also play a key role in the clearance of chronic immune reactions by providing an exit route for inflammatory cells [66]. This is important since neutrophils and other inflammatory cells invading a wound bed produce reactive oxygen species, pro-inflammatory cytokines, and proteases that can degrade extracellular matrix, impair fibroblast proliferation, and decrease re-epithelialization [24,67,68]. Indeed, work by Guc et al. suggests that the role of local lymphatic hyperplasia during wound healing is the clearance of inflammatory leukocytes [69]. In their experimental setup, bio-engineered fibrin-binding VEGF-C was used to promote local lymphatic hyperplasia and consequently facilitate wound healing in a murine wound model [69]. Interestingly, although treatment with fibrin-binding VEGF-C induced capillary lymphatic hyperplasia and the upregulation of the chemoattractant CCL21 by lymphatics, there was no change in interstitial flow with increased local lymphangiogenesis [69]. This may be attributed to the finding that lymphatic collectors were not changed by fibrin-binding VEGF-C, leaving the possibility that the clearance of interstitial fluid by lymphatics is an critical component of wound healing that requires additional study [69].

In the current study, we modified a tail model of lymphedema to create a chronic wound model that closes by epithelialization over 4 weeks in wild-type mice. Importantly, we did not ligate the collecting lymphatics in the tail and maintained capillary lymphatic drainage in the tail by excising just the ventral surface of the tail. As a result, the mice did not develop lymphedema but did have delayed wound healing. Healing in this model, in contrast to other models of excisional wound repair, does not require stenting of the wound edges and resists closure by contracture due to the fibrous connections of the tail skin to the underlying skeleton. In addition, because lymphatic drainage in the tail under normal circumstances is a linear, one-way system, the tail wound model facilitates analysis of lymphatic drainage in and around the wound. Thus, microlymphangiography in the tail model enabled us to analyze lymphatic clearance around the wound. We found that lymphatic drainage around chronic wounds is impaired and that the zone of impairment is larger than the area of the wound. Our findings are consistent with previous studies analyzing lymphatic clearance around pressure ulcers, demonstrating that lymphatic vessels are susceptible to ischemia repercussion injury and free oxygen radicals [70,71].

Our group and others have shown that chronic CD4^+^ cell inflammatory reactions play a key role in impaired lymphatic function in lymphedema and obesity [36,40,72,73,74,75]. In lymphedema, the differentiation of naïve CD4+ cells to the Th2 phenotype is important for development of pathology since depletion of these cells prevents development of disease and can be used as a means of treating the disease once it has developed [36,73]. Th2 cells impair lymphatic function by multiple mechanisms, including directly impairing lymphatic endothelial cell proliferation, migration, and function, as well as indirect effects on the extracellular matrix regulating tissue fibrosis and matrix deposition [43,44]. In the current study, we found that wounds in the tail model were infiltrated by large numbers of CD4+ cells. In addition, we noted the accumulation of Gata3+ cells suggestive of Th2 differentiation. Consistent with this finding, we found that the inhibition of Th2 differentiation with IL4- or IL13-neutralizing antibodies significantly accelerated wound repair, decreased the number of infiltrating CD4+ and Gata3+ cells, and improved lymphangiogenesis around the wound. This finding is consistent with previous studies demonstrating that high levels of IL4 or IL13 impair wound healing [76,77,78].

Our study had few limitations. First, the nature of the mouse model used makes it difficult to correlate our findings to patients. We hope that this study will serve as a foundation for incorporating lymphatic modulation and Th2 cell differentiation for patients with chronic wounds. Mehrara et al. has shown that Th2 inhibition, via the monoclonal antibody blockade of IL-4 and IL-13 antibodies, has potential for improving the clinical response to lymphedema and resulting dermal dysfunction [79]. We hope that similar studies can be performed in the realm of chronic wounds as well. Second, in this study, we did not investigate the cellular and molecular effects of Th2 cytokine inhibition on LECs as was thoroughly studied in our earlier publications using in vivo and in vitro methods. Briefly, in our earlier studies, we showed that IL-4/IL-13 are potent anti-lymphangiogenic agents whose inhibition enhances LECs proliferation, migration, and tube formation in the JAK-1-mediated signaling pathway [43,44].

## 5. Conclusions

Our findings highlight the importance of lymphangiogenesis in promoting chronic wound healing while identifying a novel mechanism of inducing lymphangiogenesis with therapeutic potential. Further studies identifying how lymphatic function impacts wound healing are warranted. We hope that this model can be used as a platform for investigating additional immunomodulatory therapies involving the inhibition of other anti-lymphangiogenic T cell cytokines like IFN-γ, IL-17 on chronic wound healing with the goal of improving clinical outcomes.

## Figures and Tables

**Figure 1 biomedicines-13-01026-f001:**
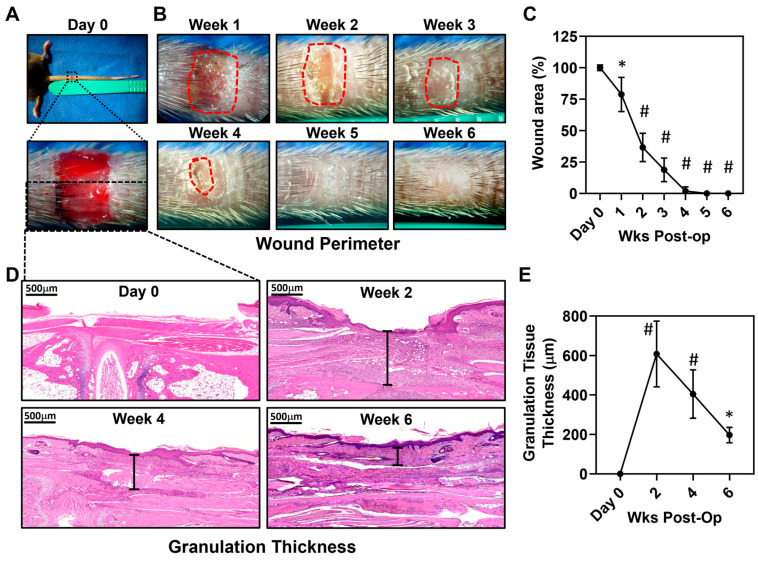
Full-thickness tail excisions result in chronic wounds that persist for 4 weeks. (**A**) Gross representative images of the full length of a wounded mouse’s tail (top) and microscopic images of the mouse’s wound (bottom) immediately following excision. (**B**) Gross representative images of mouse tail wounds during wound healing from weeks 1 to 6. The perimeter of the wounds is outlined in red. (**C**) Quantification of wound area as a percentage of the initial wound area during the time course studied. Wounds healed approximately 3–4 weeks after wounding. All wounds were fully healed by 5 weeks after wounding (*n* = 5 for all time groups, *p* < 0.05 at week 1 post-operation, *p* < 0.0001 for weeks 2–6 post-operation). (**D**) Representative high-power photo micrographs (3×) of hematoxylin and eosin staining of the wound and peri-wound areas. Examples of measurement of granulation thickness are shown through black brackets. (**E**) Quantification of granulation thickness throughout wound healing (*n* = 5/*p* < 0.0001 at weeks 2 and 4, and *p* < 0.05 at week 6 post-operation). * denotes *p* < 0.05, # denotes *p* < 0.0001.

**Figure 2 biomedicines-13-01026-f002:**
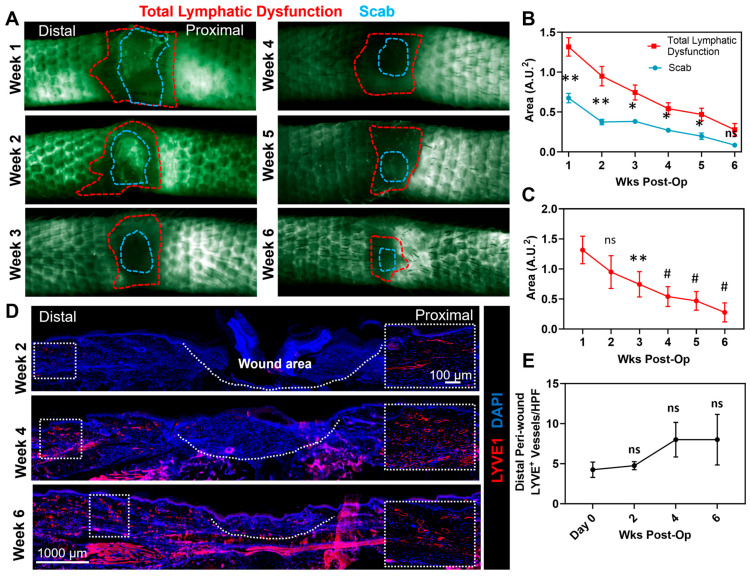
Chronic wounds cause peri-wound lymphatic dysfunction that is correlated with delayed lymphangiogenesis distal to the wound. (**A**) Representative images of FITC dextran lymphangiography in the wound and peri-wound areas. Areas of lymphatic dysfunction (red outlining) can be seen both distal and proximal to the wound bed. Scab (blue outline) was ascertained by analyzing brightfield images of the exact same location (not shown). (**B**) Quantification of the areas of total lymphatic dysfunction and the scab as wound healing occurs (*n* = 5/*p* < 0.01 at weeks 1 and 2 post-operation and *p* < 0.05 at weeks 3–5 post-operation; ns at week 6). (**C**) Quantification of the wound area as healing occurs (*n* = 5/*p* = ns at week 2, *p* < 0.01 at week 3, and *p* < 0.0001 at weeks 4–6 post-operation). (**D**) Representative high-power immunohistochemistry (zoom-out 2×; zoom-in 20×) of LYVE+ lymphatic vessel formation in the peri-wound areas. Blue is DAPI and LYVE1 is red, when the 2 co-localize it can appear magenta. (**E**) Quantification of distal peri-wound LYVE+ lymphatic vessel density during wound healing (*n* = 5, 2–3 HPF/animal, *p* = ns). * denotes *p* < 0.05, ** denotes *p* < 0.01, # denotes *p* < 0.0001.

**Figure 3 biomedicines-13-01026-f003:**
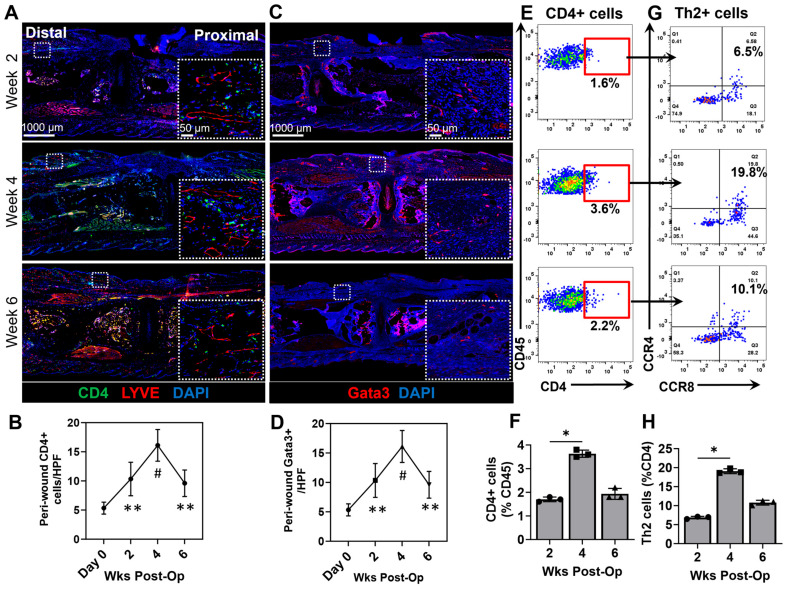
Chronic wounds result in a delayed Th2 response that persists as the wound heals. (**A**) Representative high-power immunohistochemistry (zoom-out 2×; zoom-in 40×) and quantification of (**B**) CD4+ T-helper cells in the distal peri-wound regions (*n* = 5, 2–3 HPF/animal, *p* < 0.01 at week 2, *p* < 0.0001 at week 4, and *p* < 0.001 at week 6 post-operation) (**C**) Representative high-power immunohistochemistry (zoom-out 2×; zoom-in 40×) and (**D**) quantification of Gata3+ Th2 cells in the distal peri-wound regions (*n* = 5, 2–3 HPF/animal, *p* < 0.01 at week 2, *p* < 0.0001 at week 4, and *p* < 0.001 at week 6 post-operation). (**E**) Representative flow cytometry plots displaying CD45+ on the y axis and CD4+ cells (T cells) on the x axis. (**F**) Quantification of CD4+ T cells as a percentage of CD45+ cells (*p* < 0.05 between weeks 2 and 4 post-operation). (**G**) Representative flow cytometry plots displaying CCR4+ on the y axis and CCR8+ on the x axis to identify Th2 cells. (**H**) Quantification of Th2+ T cells as a percentage of CD4+ T cells (*p* < 0.05 between weeks 2 and 4 post-operation). Red square represents gating for CD4+ & CD45+ double positive cells. As for colors, there are only 3 as mentioned in the figure: CD4 (green) LYVE (red) or Gata3 (red) DAPI (blue) and when they colocalize they may appear in different shades * denotes *p* < 0.05, ** denotes *p* < 0.01, # denotes *p* < 0.0001.

**Figure 4 biomedicines-13-01026-f004:**
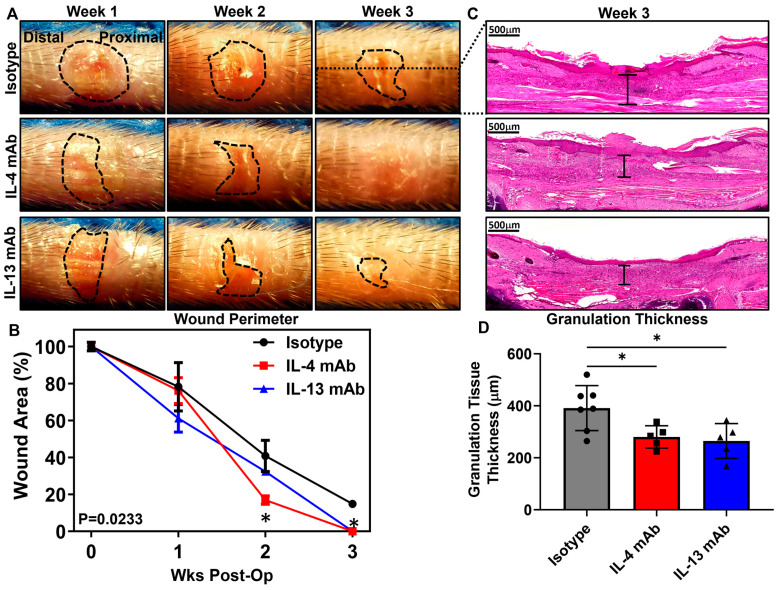
Chronic wound healing is accelerated by inhibition of Th2 cytokines. (**A**) Gross representatives images of wounds following treatment with isotype, anti-interleukin (IL)-4, or anti-IL13 monoclonal antibodies (mAb). The wound perimeter is outlined in black. Wounds treated with either IL-4 or IL-13 mAbs were closed 3 weeks after wound healing began, whereas isotype wounds remained open. (**B**) Quantification of wound area as a percentage of initial wound area during the time course studied (*n* = 5, *p* < 0.01 at weeks 2 and 3 post-surgery). (**C**) Representative high-power (3×) hematoxylin and eosin staining of the wound and peri-wound areas. Examples of measurement of granulation thickness are shown through black brackets. (**D**) Quantification of granulation thickness throughout wound healing (*n* = 5, *p* < 0.05 when comparing isotype versus IL-4 mAB and isotype versus IL-13 mAB). * denotes *p* < 0.05.

**Figure 5 biomedicines-13-01026-f005:**
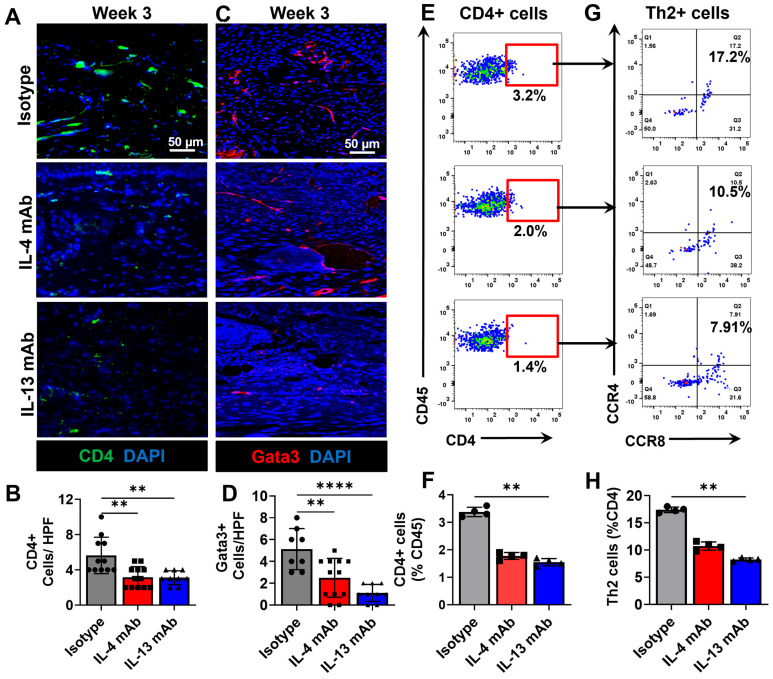
Inhibition of Th2 cytokines results in decreased peri-wound Th2 cell infiltration. (**A**) Representative high-power immunohistochemistry (zoom-out 2×; zoom-in 40×) and quantification of (**B**) CD4+ T-helper cells in the distal peri-wound regions (*n* = 5, 2–3 HPF/animal; *p* < 0.01). (**C**) Representative high-power immunohistochemistry (zoom-out 2×; zoom-in 40×) and quantification of (**D**) Gata3+ Th2 cells in the distal peri-wound regions (*n* = 5, 2–3 HPF/animal; *p* < 0.01 when comparing between isotype and IL-4 mAB and *p* < 0.0001 when comparing between isotype and IL-13 mAb-treated animals). (**E**) Representative flow cytometry plots displaying CD45+ on the y axis and CD4+ cells (T cells) on the x axis. (**F**) Quantification of CD4+ T cells as a percentage of CD45+ cells (*p* < 0.01). (**G**) Representative flow cytometry plots displaying CCR4+ on the y axis and CCR8+ on the x axis to identify Th2 cells. (**H**) Quantification of Th2+ T cells as a percentage of CD4+ T cells (*p* < 0.01). As for colors, there are only 3 as mentioned in the figure: CD4 (green) Gata3 (red) DAPI (blue) and when they colocalize they may appear in different shades. ** denotes *p* < 0.01, **** denotes *p* < 0.0001.

**Figure 6 biomedicines-13-01026-f006:**
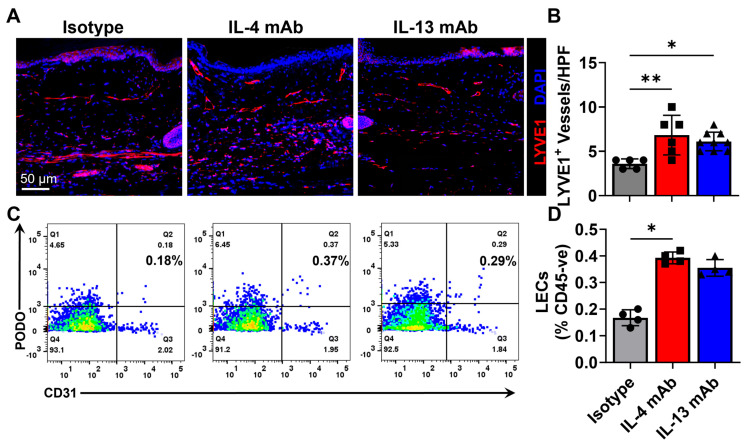
Inhibition of Th2 differentiation causes increased peri-wound lymphangiogenesis. (**A**) Representative high-power immunohistochemistry images showing LYVE-1+ lymphatic vessels (LYVE-1 is red, DAPI is blue and colocalization may result in Magenta.) and quantification of (**B**) LYVE+ lymphatic vessel formation in the peri-wound areas. (*n* = 5, 2–3 HPF/animal; *p* < 0.01 between isotype and IL-4 mAB and *p* < 0.05 between isotype and IL-13 mAb-treated animals). (**C**) Representative flow cytometry plots displaying podoplanin on the y axis and CD31+ on the x axis. (**D**) Quantification of CD31+/Podoplanin+ LECs as a percentage of CD45- cells (*p* < 0.05 between isotype and IL-4 mAb-treated animals). * denotes *p* < 0.05, ** denotes *p* < 0.01.

## Data Availability

All relevant data are within the paper and its Appendix A files.

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
