# Peer review of "Inhibition of Th2 Differentiation Accelerates Chronic Wound Healing by Facilitating Lymphangiogenesis"

_biomedicines, 2025, doi:10.3390/biomedicines13051026_

Round 1
Reviewer 1 Report
Comments and Suggestions for Authors
The article submitted to biomedicine regarding the role of Th2 lymphocyte differentiation in lymphangiogenesis and consequently in wound healing is well written and quite clear except in some parts.
For example, in the materials and methods it is not clear how to evaluate the size of the wound in relation to the creation of new lymphatic vessels. Furthermore, the bibliography of the article and blood vessels is not particularly updated. For example, we suggest to cite and discuss a review from 2023 (Breaking a Vicious Circle: Lymphangiogenesis as a New Therapeutic Target in Wound Healing. Biomedicines. 2023 Feb 21;11(3):656). There are also several typos.
Finally, it is essential to explain why the chosen wound model is considered chronic, since it heals spontaneously in 4/5 weeks because of its particular size and location at the tail level.
Author Response
Reviewer 1:
The article submitted to biomedicine regarding the role of Th2 lymphocyte differentiation in lymphangiogenesis and consequently in wound healing is well written and quite clear except in some parts.
For example, in the materials and methods it is not clear how to evaluate the size of the wound in relation to the creation of new lymphatic vessels.
We thank the reviewer for the positive comments for our manuscript.
Size of the wound is defined by the distance from the proximal end of the wound bed to the distal end. Peri-wound area is defined as area outside or around the wound with in 3mm either distal or proximal to the wound bed. Analysis of new lymphatic vessels was performed within the peri-wound area in using 20x view high power field images from each group. These details were clarified in the revised manuscript and can be found on page 3, lines 102-107.
Furthermore, the bibliography of the article and blood vessels is not particularly updated. For example, we suggest citing and discuss a review from 2023 (Breaking a Vicious Circle: Lymphangiogenesis as a New Therapeutic Target in Wound Healing. Biomedicines. 2023 Feb 21;11(3):656). There are also several typos.
The specific publication is now cited and discussed in the revised manuscript. Page 2, line 48, citation #31.
Thank you, we have reviewed the manuscript and ensured there are no typos.
Finally, it is essential to explain why the chosen wound model is considered chronic, since it heals spontaneously in 4/5 weeks because of its particular size and location at the tail level.
In human data and studies there are multiple timepoints chosen to define “chronic” wounds: as wounds that fail to close in a range of 4-12+ weeks. For the purpose of our study, and the mouse model used, the wounds failure to close before 4-5 weeks is considered chronic as mice heal more quickly than humans, and to our knowledge before this study, no model had shown a wound remaining open for this length of time.
Reviewer 2 Report
Comments and Suggestions for Authors
This manuscript successfully simulates chronic wounds in a mouse tail skin model and demonstrates accelerated wound healing and lymphatic vessel regeneration through the inhibition of Th2 cell differentiation, showcasing a degree of innovation. However, the study has notable limitations in several areas, including the functional improvement of newly formed lymphatic vessels, exploration of molecular mechanisms, assessment of biosafety, and potential impacts on other tissue functions. It is recommended that the authors address these concerns by revising the manuscript extensively, incorporating additional experiments and discussions to enhance the overall quality and academic value of the research.
- In the animal experiments where Th2 cell differentiation was inhibited by neutralizing antibodies, the study focused solely on the increase in the number of newly formed lymphatic vessels. However, it was previously mentioned that some of these newly formed lymphatic vessels in chronic wounds are dysfunctional and unable to effectively transport interstitial fluid. Therefore, does the inhibition of Th2 cell differentiation lead to an improvement in the drainage function of these newly formed lymphatic vessels? Could the authors consider incorporating imaging techniques to further investigate changes in the drainage function of these vessels?
- Could the experiment involving the inhibition of Th2 cell differentiation by neutralizing antibodies potentially affect angiogenesis or the regeneration of other tissues around the wound? This aspect may require further evaluation.
- Does the inhibition of Th2 differentiation through the neutralization of IL-4 and IL-13 antibodies pose any additional adverse effects on the immune system, such as immunosuppression or an increased risk of infection? Have these potential risks been adequately assessed?
- In Figure 2A, the lymphatic drainage contrast image at the 2-week time point shows a relatively prominent bright spot in the central part of the wound. What is the underlying reason for this observation?
- The manuscript relies solely on animal models to study the effects of Th2 cell inhibition on chronic wound healing and lymphatic vessel regeneration, without further validation of the molecular mechanisms through cell-based experiments. Could the authors consider supplementing their findings with cell experiments to verify these mechanisms? Additionally, could this inhibition affect cell types other than lymphatic endothelial cells (LECs)?
- The proportion of references from the last five years is relatively low (approximately 13%), which may result in the research background and discussion sections lacking up-to-date information. It is recommended that the authors update their references, particularly by including relevant studies from the past three to five years, to enhance the timeliness and relevance of their research.
Minior Concerns
- The resolution of some figures (e.g., Figure 1 and Figure 2) is suboptimal, and the color contrast is insufficient, which compromises the clarity of the data. For instance, the scale bar in Figure 1D is difficult to discern and requires improvement.
- The introduction section contains formatting issues. For example: “However, although the lymphatic system is a key regulator of immune cell trafficking and clearance of cellular debris; two processes that are important for chronic wound healing to occur [20-22]; the contribution of this system to wound healing remains understudied. . Therefore, the goal of this current study was twofold.” The punctuation and sentence structure need to be revised for clarity and coherence.
- For Figure 4A, it would be beneficial to provide images of additional samples to allow for a more comprehensive evaluation.
- IL-4 and IL-13 are well-studied cytokines, and the study lacks exploration of novel molecular mechanisms. Additionally, have the authors considered investigating other factors in future research?
- In the animal experiments where Th2 differentiation was inhibited via intraperitoneal injection of monoclonal antibodies targeting IL-4 and IL-13, the potential systemic toxicity of the injected agents to major organs (e.g., heart, liver, spleen, lungs, kidneys) was not evaluated. Were any biosafety experiments conducted to assess this?
- The discussion section should be expanded to include a critical evaluation of the study's limitations, as well as propose directions for future research and potential clinical applications.
Author Response
Reviewer 2:
This manuscript successfully simulates chronic wounds in a mouse tail skin model and demonstrates accelerated wound healing and lymphatic vessel regeneration through the inhibition of Th2 cell differentiation, showcasing a degree of innovation. However, the study has notable limitations in several areas, including the functional improvement of newly formed lymphatic vessels, exploration of molecular mechanisms, assessment of biosafety, and potential impacts on other tissue functions. It is recommended that the authors address these concerns by revising the manuscript extensively, incorporating additional experiments and discussions to enhance the overall quality and academic value of the research.
We thank the reviewer for the critical comments. We have tried to address the reviewer’s concerns below.
- In the animal experiments where Th2 cell differentiation was inhibited by neutralizing antibodies, the study focused solely on the increase in the number of newly formed lymphatic vessels. However, it was previously mentioned that some of these newly formed lymphatic vessels in chronic wounds are dysfunctional and unable to effectively transport interstitial fluid. Therefore, does the inhibition of Th2 cell differentiation lead to an improvement in the drainage function of these newly formed lymphatic vessels? Could the authors consider incorporating imaging techniques to further investigate changes in the drainage function of these vessels?
Thank you for this comment. Investigating the function of the newly formed vessels is a good idea and has been done in our previous work in which they investigated lymphatic function following Th2 inhibition in mouse model of lymphedema which is a more severe model than the wound model we used in this manuscript. This was done on mice in which the lymphatics were surgically ligated and functional improvement was observed.1 in this manuscript, our model of a chronic wound, the wound is more superficial than previously described, and the lymphatics were not ligated. Thus, we expect that we would see improvement in our model, and it would be redundant to repeat the same.
Moreover, to perform the functional experiment, we would need to we repeat our entire experiment which would be cost prohibitive as well as a waste of animal life to prove something that has been already done. If we did this, we would only gain one piece of data, which would serve as a confirmation for our other data and therefore has limited utility for the context of our study. We would possibly do this in future studies where we can induce chronic wounds on other areas of the mouse (such as the hindlimb), in which we could more accurately measure lymphatic function.
- Avraham T, Zampell JC, Yan A, et al. Th2 differentiation is necessary for soft tissue fibrosis and lymphatic dysfunction resulting from lymphedema. FASEB J. 2013;27(3):1114-1126. doi:10.1096/fj.12-222695
- Could the experiment involving the inhibition of Th2 cell differentiation by neutralizing antibodies potentially affect angiogenesis or the regeneration of other tissues around the wound? This aspect may require further evaluation.
We thank the reviewer for this important question. Interestingly, we didn’t observe any changes in angiogenesis upon Th2 cytokine inhibition in our model. In figure 6C, flowcytometry, we show that there were no changes in single positive CD31 cells (representing blood endothelial cells; BECs) upon inhibition of Th2 cell differentiation, indicating no differences in BECs or angiogenesis. Now we have added flowcytometry as well immuno-histological quantification of CD31+ BECs in the supplementary figure 1A-D.
- Does the inhibition of Th2 differentiation through the neutralization of IL-4 and IL-13 antibodies pose any additional adverse effects on the immune system, such as immunosuppression or an increased risk of infection? Have these potential risks been adequately assessed?
Thank you for this critical question. We have previously studied inhibition of Th2 differentiation in mouse models as well as a clinical trial. In both the human and mouse data, we have not seen any adverse effects of Th2 inhibition in regard to immunosuppression or increased susceptibility to infections. 1, 2 We have no reason to believe that such immunosuppressive changes would occur in this model.
- Avraham T, Zampell JC, Yan A, et al. Th2 differentiation is necessary for soft tissue fibrosis and lymphatic dysfunction resulting from lymphedema. FASEB J. 2013;27(3):1114-1126. doi:10.1096/fj.12-222695
- Mehrara BJ, Park HJ, Kataru RP, et al. Pilot Study of Anti-Th2 Immunotherapy for the Treatment of Breast Cancer-Related Upper Extremity Lymphedema. Biology (Basel). 2021;10(9):934. Published 2021 Sep 18. doi:10.3390/biology10090934
- In Figure 2A, the lymphatic drainage contrast image at the 2-week time point shows a relatively prominent bright spot in the central part of the wound. What is the underlying reason for this observation?
In our experience with this imaging techniques, scabs and wounds sometimes have autofluorescence. In Figure 2A the bright spot shows the scab and its autofluorescence.
- The manuscript relies solely on animal models to study the effects of Th2 cell inhibition on chronic wound healing and lymphatic vessel regeneration, without further validation of the molecular mechanisms through cell-based experiments. Could the authors consider supplementing their findings with cell experiments to verify these mechanisms? Additionally, could this inhibition affect cell types other than lymphatic endothelial cells (LECs)?
The goal of our study was to investigate the effects of Th2 cell inhibition on in-vivo models, with the eventual goal to relate our findings clinically. We focused on lymphatic growth in relation to wound healing, and we did not aim to study the underlying cellular and molecular pathophysiology in this study. Because our earlier in-vitro and in-vivo based research we have thoroughly studied the effects of Th2 inhibition on LECs marker gene expression, putative signaling pathways involved, growth, proliferation, apoptosis, tube formation, migration, and other cellular, molecular aspects 1,2. Thus, studying the molecular mechanisms would be repetitive and is out of the scope of the current study.
In regard to your question about the effect of Th2 inhibition on cells other than LECs, this study found no effect on blood endothelial cells, as evidenced by stable CD31+ cells across groups (Supplementary Fig 1). Previous work by our lab has shown that inhibition of Th2 differentiation results in decreased levels of both Th1 and Th2 cells.3
- Savetsky, I. L., Ghanta, S., Gardenier, J. C., Torrisi, J. S., Garcia Nores, G. D., Hespe, G. E., Nitti, M. D., Kataru, R. P., & Mehrara, B. J. (2015). Th2 cytokines inhibit lymphangiogenesis. PLoS One, 10(6), e0126908. https://doi.org/10.1371/journal.pone.0126908
- Shin, K., Kataru, R. P., Park, H. J., Kwon, B. I., Kim, T. W., Hong, Y. K., & Lee, S. H. (2015). TH2 cells and their cytokines regulate formation and function of lymphatic vessels. Nat Commun, 6, 6196. https://doi.org/10.1038/ncomms7196
- Avraham T, Zampell JC, Yan A, et al. Th2 differentiation is necessary for soft tissue fibrosis and lymphatic dysfunction resulting from lymphedema. FASEB J. 2013;27(3):1114-1126. doi:10.1096/fj.12-222695
- The proportion of references from the last five years is relatively low (approximately 13%), which may result in the research background and discussion sections lacking up-to-date information. It is recommended that the authors update their references, particularly by including relevant studies from the past three to five years, to enhance the timeliness and relevance of their research.
Thank you for pointing this out. We have updated the references and bibliography accordingly, and new references from past 5 years now comprise approximately 30% of the total.
Minor Concerns
- The resolution of some figures (e.g., Figure 1 and Figure 2) is suboptimal, and the color contrast is insufficient, which compromises the clarity of the data. For instance, the scale bar in Figure 1D is difficult to discern and requires improvement.
Thank you, we have improved the resolution and enhanced the visibility of the scale bars in both 1D and 4C and wherever appropriate. We have also ensured that the images are at 300 DPI or more resolution.
- The introduction section contains formatting issues. For example: “However, although the lymphatic system is a key regulator of immune cell trafficking and clearance of cellular debris; two processes that are important for chronic wound healing to occur [20-22]; the contribution of this system to wound healing remains understudied. Therefore, the goal of this current study was twofold.” The punctuation and sentence structure need to be revised for clarity and coherence.
Thank you, this has been revised on page 2, line 47-50.
- For Figure 4A, it would be beneficial to provide images of additional samples to allow for a more comprehensive evaluation.
The data was compiled using all the available samples and images and is comprehensive with the representative images used in the figure.
- IL-4 and IL-13 are well-studied cytokines, and the study lacks exploration of novel molecular mechanisms. Additionally, have the authors considered investigating other factors in future research?
As mentioned earlier the molecular mechanism of action of IL-4/13 inhibition of LECs is thoroughly studied in our earlier research. Yes, we would like to consider inhibiting other T cell secreted cytokines like IL-17 on lymphangiogenesis and would healing in our future research.
- In the animal experiments where Th2 differentiation was inhibited via intraperitoneal injection of monoclonal antibodies targeting IL-4 and IL-13, the potential systemic toxicity of the injected agents to major organs (e.g., heart, liver, spleen, lungs, kidneys) was not evaluated. Were any biosafety experiments conducted to assess this?
We have not noticed any visible symptoms of sickness in mice. We have previously used these antibodies and found that no side effects were observed with this dosage and regimen of the neutralizing antibody treatment.1,2
- Avraham T, Zampell JC, Yan A, et al. Th2 differentiation is necessary for soft tissue fibrosis and lymphatic dysfunction resulting from lymphedema. FASEB J. 2013;27(3):1114-1126. doi:10.1096/fj.12-222695
- Shin, K., Kataru, R. P., Park, H. J., Kwon, B. I., Kim, T. W., Hong, Y. K., & Lee, S. H. (2015). TH2 cells and their cytokines regulate formation and function of lymphatic vessels. Nat Commun, 6, 6196. https://doi.org/10.1038/ncomms7196
- The discussion section should be expanded to include a critical evaluation of the study's limitations, as well as proposed directions for future research and potential clinical applications.
This has been added to the discussion and can be found on page 12, lines 402-415.
Reviewer 3 Report
Comments and Suggestions for Authors
This is a really interesting manuscript with novel findings, clear presentation, and high relevance.
The manuscript is well-written, methodologically rigorous, and highly relevant to basic science and clinical audiences.
I have only two suggestions for improving the quality of the paper.
- I recommend enhancing the translational message by briefly discussing potential implications or limitations concerning human applicability in the discussion.
- Consider emphasizing in the conclusion how this model can act as a platform for testing additional immunomodulatory therapies.
Author Response
Reviewer 3:
This is an interesting manuscript with novel findings, clear presentation, and high relevance.
The manuscript is well-written, methodologically rigorous, and highly relevant to basic science and clinical audiences.
I have only two suggestions for improving the quality of the paper.
I recommend enhancing the translational message by briefly discussing potential implications or limitations concerning human applicability in the discussion.
Thank you for your constructive criticism. We have added the above points to the manuscript, which can be found on lines 402-415.
Consider emphasizing in the conclusion how this model can act as a platform for testing additional immunomodulatory therapies.
This has been added to the manuscript, lines 420-423.
Round 2
Reviewer 1 Report
Comments and Suggestions for Authors
The article by Pollak et al. has been modified to clarify some technical and substantial aspects, such as the concept of chronic wound in the proposed murine model. Therefore, in this new form, the article is certainly worthy of being published on the Biomedicine special issue.
Reviewer 2 Report
Comments and Suggestions for Authors
I have no more question.